# Age- and Sex-Dependent Behavioral and Neurochemical Alterations in hLRRK2-G2019S BAC Mice

**DOI:** 10.3390/biom13010051

**Published:** 2022-12-27

**Authors:** Ning Yao, Olga Skiteva, Karima Chergui

**Affiliations:** Molecular Neurophysiology Laboratory, Department of Physiology and Pharmacology, Karolinska Institutet, 171 64 Stockholm, Sweden

**Keywords:** BAC: bacterial artificial chromosome, DA: dopaminergic, DAT: dopamine transporter, KI: knock-in, LRRK2: leucine-rich repeat kinase 2, PD: Parkinson’s disease, SNc: substantia nigra pars compacta, TH: tyrosine hydroxylase

## Abstract

The G2019S mutation in the leucine-rich repeat kinase 2 (*LRRK2*) gene is associated with late-onset Parkinson’s disease (PD). Although PD affects men and women differently, longitudinal studies examining sex- and age-dependent alterations in mice carrying the G2019S mutation are limited. We examined if behavioral and neurochemical dysfunctions, as well as neurodegeneration, occur in male and female BAC LRRK2-hG2019S (G2019S) mice, compared to their age-matched wild type littermates, at four age ranges. In the open field test, hyperlocomotion was observed in 10–12 month old male and 2–4.5 months old female G2019S mice. In the pole test, motor coordination was impaired in male G2019S mice from 15 months of age and in 20–21 months old female G2019S mice. In the striatum of G2019S male and female mice, the amounts of tyrosine hydroxylase (TH), measured with Western blotting, were unaltered. However, we found a decreased expression of the dopamine transporter in 20–21 month old male G2019S mice. The number of TH-positive neurons in the substantia nigra compacta was unaltered in 20–21 month old male and female G2019S mice. These results identify sex- and age-dependent differences in the occurrence of motor and neurochemical deficits in BAC LRRK2-hG2019S mice, and no degeneration of DA neurons.

## 1. Introduction

Parkinson’s disease (PD) is a progressive neurodegenerative disease that causes several motor and non-motor impairments, in particular, bradykinesia, rigidity, and tremor, the three cardinal motor symptoms of PD [1]. In PD, dopaminergic (DA) neurons in the substantia nigra compacta (SNc), as well as other neuronal populations, degenerate progressively. This loss causes a dramatic reduction in the content of dopamine in the caudate and putamen/striatum, which receive a dense innervation from SNc-DA neurons. It is suggested that axons of these DA neurons degenerate first, and that DA cell bodies in the SNc degenerate later, hence the “dying back” process [2]. Indeed, clinical motor signs appear when around 30–50% of SNc neurons and 80% of striatal dopamine are lost, and motor disturbances progressively worsen [1,3]. Most cases of PD are sporadic, but mutations of specific genes occur in around 5 % of PD patients, causing early- or late-onset PD [4]. Mutations in the leucine-rich repeat kinase 2 gene (*LRRK2*, *PARK8*, encoding dardarin protein) are among the most common causes of familial PD and produce autosomal dominant late-onset PD that is similar to idiopathic PD. The G2019S point mutation in the *LRRK2* gene is a common, and the most studied, pathogenic mutation [5,6,7]. This mutation might increase the susceptibility of DA neurons to degeneration. Interestingly, in a recent longitudinal clinical study with carriers of the G2019S mutation, which do not manifest symptoms, 13% of the patients had a significant reduction in the dopamine transporter (DAT) in the caudate putamen [8], which indicates neurochemical alterations before the disease develops. In order to investigate the roles of LRRK2, and mutated LRRK2, in the development of Parkinsonism, several rodent models bearing the G2019S mutation have been generated using, for example, bacterial artificial chromosome (BAC) and knock-in (KI) to express human or murine LRRK2 or LRRK2-G2019S [9,10,11,12,13,14,15]. However, most studies use male mice, and no longitudinal studies have been performed to examine the influence of sex and age in behavioral and neurochemical alterations linked to the G2019S mutation. Sex is an important variable in such analyses in animal models of PD because the clinical features of PD differ between men and women, and men are 1.5–2 times more likely to develop PD than women, suggesting distinct pathophysiological mechanisms underlying the disease [16]. In addition, DA neurons in female rodents are less vulnerable to factors inducing degeneration than male rodents [16]. In the present study, we examined if mice carrying the G2019S mutation in the *LRRK2* gene (G2019S mice), which were generated via BAC transgenesis of human LRRK2, display age- and sex-dependent occurrence of PD-like motor symptoms, changes in DA markers, and neurodegeneration.

## 2. Materials and Methods

### 2.1. Animals

Animal experiments were approved by our local ethical committee (Stockholms norra djurförsöksetiska nämnd, 20464-2020). We used mice that express the G2019S mutation in the human *LRRK2* gene, generated via BAC transgenesis. These mice were obtained from The Jackson Laboratory (C57BL/6J-Tg(LRRK2-G2019S)2AMjff/J, JAX stock #018785; RRID:IMSR_JAX:018785), were bred in our animal facility, and were mated as Noncarrier × Hemizygote [17]. We used male and female hemizygous mice (G2019S) and non-transgenic wildtype (WT) littermates at four different age ranges: 2–4.5 months, 10–12 months, 15 months, and 20–21 months. Due to the low availability of 20–21-month-old mice, the number of animals tested in this age group is lower than in the other groups. All studied mice were backcrossed on a C57BL/6J genetic background. Mice were housed in small groups (2–5 per cage, IVC Mouse–GM500) in a humidity-controlled room with a 12:12 h light/dark cycle and had free access to food and water.

### 2.2. Behavioral Tests

Mice were allowed to adapt to the testing room for at least 30 min before the behavioral tests were conducted. In the open field test, mice were placed individually in the center of an open field arena (46 cm × 46 cm) and allowed to move freely for 60 min. Horizontal locomotor activity was tracked with a ceiling camera coupled to the EthoVision XT11.5 (Noldus, Wageningen, The Netherlands) software. We measured the total distance covered in the arena during the 60 min trial period. We assessed fine motor behavior by performing the pole test, in which mice were placed individually on top of a vertical pole (diameter: 8 mm, height: 50 cm) with their head facing upwards. During the first two days of the test, mice were trained to turn and descend the pole back into a cage. On the third day of the test, mice were videotaped while descending the pole for a total of three trials. The time taken by the mice to turn downward (Tturn) and the total time to descend the pole (Ttotal) were measured manually with a timer. Data represents the average of three trials.

### 2.3. Western Blotting

The dorsal striatum was dissected from the fresh brain slices made with a microslicer (VT 1000S, Leica Microsystem, Wetzlar, Germany) in artificial cerebrospinal fluid containing (in mM): NaCl (126), KCl (2.5), NaH_2_PO_4_ (1.2), MgCl_2_ (1.3), CaCl_2_ (2.4), glucose (10), and NaHCO_3_ (26), frozen and stored at −80 °C until processed. The samples were sonicated in 1% sodium dodecyl sulfate (SDS) and boiled for 10 min. The 1 % SDS was diluted in water from 10% SDS (prepared with 18 MΩ water, Bio-Rad, Cat. No. 1610416, Hercules, CA, USA). Protein concentration was determined in each sample with a bicinchoninic acid protein assay (BCA-kit, Pierce, Rockford, IL, USA, Cat. No. 23225). Equal amounts of protein (30 μg) were re-suspended in the sample buffer (4 × Laemmli Sample Buffer, Bio-Rad, Hercules, CA, USA, Cat. No. 1610747, added with 10 % β-mercaptoethanol, Sigma, St Louis, MO, USA, Cat. No. M3148) and separated by SDS–polyacrylamide gel electrophoresis using a 9 % acrylamide gel (Acrylamide/Bis-acrylamide, 30% solution: Sigma, St Louis, MO, USA, Cat. No. A3699) and transferred to a nitrocellulose transfer membrane (Bio-Rad, Hercules, CA, USA, Cat. No. 1620115). We performed Western blotting in duplicates for the samples from aged mice (20–21 months), as performed in previous studies [18], due to the low availability of mice in this group. The membranes were incubated for 1 h at room temperature with 5% (*w*/*v*) fat-free dry milk (Cell Signaling, Danvers, MA, USA, Cat. No. 9999S) in TBS-T (Tris base 0.05 mol/L, NaCl 0.15 mol/L, tween 0.1%). Immunoblotting was carried out with primary antibodies in 5% dry milk dissolved in TBS-T at 4 °C overnight. Antibodies were obtained from Sigma-Aldrich, St Louis, MO, USA (tyrosine hydroxylase (TH), Cat. No. T2928, dilution 1:2000; β-actin, Cat. No. A2228, dilution 1:2000) and Millipore, Temecula, CA, USA (DAT, Cat. No. MAB369, dilution 1:1000). The membranes were washed three times with TBS-T and incubated for 1 h at room temperature, with secondary horseradish peroxidase-linked Anti-Rat IgG (H + L) (Thermo Scientific, Rockford, IL, USA, Cat. No. 31470, 1:5000 dilution) or Anti-Mouse IgG (H + L) (Thermo Scientific, Rockford, IL, USA, Cat. No. 32230, 1:5000 dilution). At the end of the incubation, membranes were washed six times with TBS-T, and immunoreactive bands were detected by enhanced chemiluminescence (Bio-Rad, Hercules, CA, USA, Cat. No. 170-5061). The membranes were then scanned in ChemiDoc MP system (Bio-Rad, Hercules, CA, USA) and quantified with ImageJ 1.50b software (NIH, Bethesda, MD, USA). The protein amounts were normalized to the value of β-actin and expressed as a percentage of the averaged value obtained for WT mice.

### 2.4. Immunofluorescence and Cell Count

Mouse brain tissues containing the midbrain were freshly dissected and post-fixated over 2 nights in 4% paraformaldehyde (Sigma-Aldrich, St Louis, MO, USA, Cat. No. 16005) in phosphate buffer saline (PBS; Sigma-Aldrich, St Louis, MO, USA, Cat. No. P4417) and dehydrated in 30% sucrose-PBS buffer for 2–3 days. Dehydrated brains were embedded in OCT cryomount (Cat. No. 45830, Histolab, Gothenburg, Sweden), frozen at −20 °C, and sliced with a MICROM cryostat (HM 500 M) at a 40 μm thickness. The sections were collected and stored in NaN_3_ (0.01% in PBS) in 24-well plates at 4 °C. Free floating brain sections containing the midbrain were incubated for one night at 4 °C in a TH primary antibody (Millipore, Temecula, CA, USA, Cat. No. AB152, dilution 1:2000). Sections were washed 3 times in PBS and incubated in Alexa Fluor^®^ 488-conjugated goat anti-rabbit-IgG (A-11004, Thermo Scientific, Rockford, IL, USA, dilution 1:2000) for 2 h at room temperature, followed by re-washing in PBS and mounting with 70% glycerol. The sections were imaged on a Carl Zeiss LSM 880 confocal microscope (Oberkochen, Germany) using a 20x objective. Images were z-stacked. For counting the number of TH-positive cells, TH immunostaining was performed in 5 sections containing different antero-posterior (Bregma −4.80–6.04) regions of the SNc for each mouse examined. After confocal scanning, the number of TH-positive cells in the SNc in both sides of each section was counted manually and blindly by another researcher using Cell Counter plugin in Fiji [19] and a surface cell count method described earlier [20,21]. The number of cells from the two sides of the same section were added, and the average of all sections was calculated for each mouse.

### 2.5. Statistical Analysis

The GraphPad Prism 9 software was used for data analysis and statistics. Data is expressed as mean ± S.E.M. with *N* indicating the number of mice tested. We used the Shapiro-Wilk test to assess the normal distribution of the data. The statistical significance of the results was assessed using the Student’s *t*-test for unpaired observations when datasets fulfilled normal distribution. The Mann-Whitney U test was used when non-normal distributed datasets were tested. All tests were two-tailed. Significant levels were set at *p* < 0.05.

## 3. Results

### 3.1. Male and Female G2019S Mice Display Hyperlocomotion at Different Ages

Male and female WT and G2019S mice were examined for possible locomotor impairment in the open field test at four different age ranges: 2–4.5 months (young adult), 10–12 months (middle-aged), 15 months (upper limit of middle age), and 20–21 months (old) [22]. Male G2019S mice aged 10–12 months displayed hyperlocomotion as shown by a significant increase in the total distance covered in the arena during the 60 min duration of the test (Figure 1A). Hyperlocomotion was also observed in female G2019S mice but at a younger age range (2–4.5 months, Figure 1B). Locomotion was unaltered in the other age ranges in both male and female mice.

### 3.2. Male and Female G2019S Mice Display Fine Motor Impairment at Different Ages

We assessed motor coordination with the pole test. Male and female G2019S mice aged 2–4.5 months and 10–12 months performed comparatively to their WT littermates (Figure 2). At 15 months, male G2019S mice displayed an increased time to turn downward (Tturn) from the top of the pole, which persisted in 20–21 months old mice (Figure 2A). Female G2019S mice also showed increased Tturn, but only at 20–21 months of age (Figure 2B). The total time taken to descend the pole (Ttotal) was significantly increased in both male and female 20–21 months old G2019S mice (Figure 2). These results show that male and female G2019S mice display impaired fine motor behavior as they become old and that these deficits occur later in female mice compared to male G2019S mice, demonstrating the slow, progressive, sex-dependent onset of motor deficits.

### 3.3. Old Male G2019S Mice Display Decreased DAT Amount in the Striatum

In PD, the axon terminals of SNc-DA neurons degenerate first, leading to decreased DA in the striatum as well as DA markers, such as TH (the rate limiting enzyme in the synthesis of dopamine), and DAT. To investigate whether DA deficits occur in G2019S mice and whether these deficits are associated with motor impairments, we performed Western blotting experiments and measured the amounts of TH and DAT in the striatum of G2019S and WT mice. We performed these experiments at two age ranges: 10–12 and 20–21 month old mice when motor coordination was unaltered and when it was impaired, respectively. We found that TH amounts were unchanged at these two age ranges in male and female G2019S mice (Figure 3). However, 20–21 month old male, but not female, G2019S mice had reduced striatal amounts of DAT (Figure 3).

### 3.4. Intact Cell Counts in the SNc of Old WT and G2019S Male and Female Mice

We previously demonstrated that 10–12 month old G2019S mice had similar numbers of TH-positive neurons in the SNc as WT mice of the same age [17]. We asked if degeneration of SNc-DA neurons cell bodies occurred in older G2019S mice. We performed immunohistochemistry of TH and counted the number of TH-positive neurons in the SNc of 20–21 months old WT and G2019S male and female mice. We found that the numbers of TH-positive neurons in the SNc were similar in WT and G2019S mice (Figure 4), demonstrating a lack of SNc-DA neuron loss in old mice.

## 4. Discussion

We report the results of a longitudinal study that identifies age- and sex-related behavioral and neurochemical alterations in mice carrying the G2019S mutation in the *LRRK2* gene (G2019S mice), generated via BAC transgenesis of human LRRK2. Using the same mouse line, we recently demonstrated that fine motor coordination and balance were unaltered in 10–12 month old G2019S mice, but that these mice displayed an increased locomotion and exploratory behavior compared to their WT littermates [17]. In the present study, we confirm that both male and female G2019S mice display hyperlocomotion (i.e., an increased total distance covered in the open field for 60 min). However, this occurs at different ages in male mice (10–12 months) and female mice (2–4.5 months). In older mice, hyperlocomotion was not permanent because locomotion normalized compared to WT mice. We also found that deficits in motor coordination occur in both male and female G2019S mice at an advanced age, but that male mice start developing such impairments before female mice. Altered motor behaviors were also described in other BAC and KI LRRK2-G2019S mice, in which hyperlocomotion was shown to be due to enhanced LRRK2 kinase activity and was associated with an increased DAT amount in the striatum [18,23,24,25,26]. In the present study, we did not observe any significant difference in the amounts of DAT in the striatum of male and female G2019S mice compared to WT mice. Hyperlocomotion, therefore, seems to be independent of changes in the amounts of DAT, as well as TH, in the striatum of the G2019S mice used in the present study. Other mechanisms might be involved in hyperlocomotion at different time points in male and female mice. Such mechanisms might include an increased dopamine content in the striatum, an increased dopamine release probability, and an increased dopamine D1 receptor expression, as shown in other PD mouse models [27,28]. Our study demonstrates a decrease in striatal DAT in 20–21 month old male, but not female, G2019S mice. Although the amounts of TH were unaltered in old mice, the decrease in striatal DAT might indicate the degeneration of DA axon terminals. We found that this decrease is associated with motor deficits in male G2019S mice, suggesting that this altered behavior might be due to the loss of DA axon terminals in the striatum. Remarkably, no decrease in striatal DAT or TH was observed in old female G2019S mice despite the occurrence of motor deficits in this group. The mechanisms underlying motor impairments in old female G2019S mice might thus differ from those in male mice. Possible mechanisms include, for example, an altered ability of DA axon terminals to release dopamine in the striatum and an impaired glutamatergic synaptic transmission and plasticity in the striatum, as has been demonstrated in previous studies [26]. In addition, future studies employing transcriptomic and/or proteomic tools, as previously conducted [29], to analyze SNc-DA neurons in male and female BAC G2019S mice at different ages, could identify possible mechanisms involved in the observed impairments.

Few studies have demonstrated a loss of DA neurons linked to increased LRRK2 kinase function associated with the G2019S mutation. Thus, aged mice overexpressing hG2019S under the PDGF-β promoter demonstrated a significant DA neuron loss [13,30]. Using another methodological approach, whereby the expression of LRRK2-G2019S was restricted to DA neurons, Xiong et al. (2018) reported a kinase-dependent DA neuron degeneration [31]. The present work demonstrates that, in our mouse line, DA neurons in the SNc of old mice do not degenerate. These results confirm the lack of degeneration reported by previous studies using other BAC G2019S mice and rats, as well as KI G2019S mice [11,12,32]. These models are, however, more susceptible to toxins such as 1-methyl-4-phenyl-1,2,5,6-tetrahydropyridine (MPTP) [33,34]. Together, and as discussed in earlier work [9,23], these observations confirm that G2019S mice generated through different methods display diverse phenotypes. Accordingly, another line of hLRRK2-G2019S BAC mice was shown in previous studies to display impaired DA neurotransmission and neurogenesis, cognitive impairment, behavioral alterations, and neuronal loss [11,35,36,37]. Different backgrounds of the mice used to generate the different lines, as well as sex-related differences, likely contribute to the variability in the results obtained with different lines of mice.

Rodent models of PD linked to the G2019S mutation in the *LRRK2* gene do not recapitulate all the pathological features of the disease. Such features include the sex-specific, age-dependent, slow development of motor, and neurochemical, changes. Indeed, pathophysiological mechanisms underlying PD might differ between men and women. In addition, in patients with sporadic late-onset PD, the age of onset of motor symptoms is around 62 years, which is equivalent to the age of the “old” group of mice used in this study (20–21 months). Our longitudinal study demonstrates that both male and female mice show slowly developing impairment in fine motor control, but that a significant decrease in striatal DAT amounts is observed only in old G2019S male mice. Our study points to the need to analyze symptomatic mice at an old age (i.e., at least 20 months) for motor and neurochemical alterations. It also demonstrates the importance of considering the sex of the mice because male and female mice develop motor alterations at different ages, and female mice do not show altered striatal DAT. The BAC LRRK2-hG2019S model used in the present study is well suited to decipher cellular dysfunctions in DA and non-DA neurons, which occur before the onset of PD-like motor impairments [17,38], but not to address the mechanisms of the degeneration of SNc-DA neurons somata.

## Figures and Tables

**Figure 1 biomolecules-13-00051-f001:**
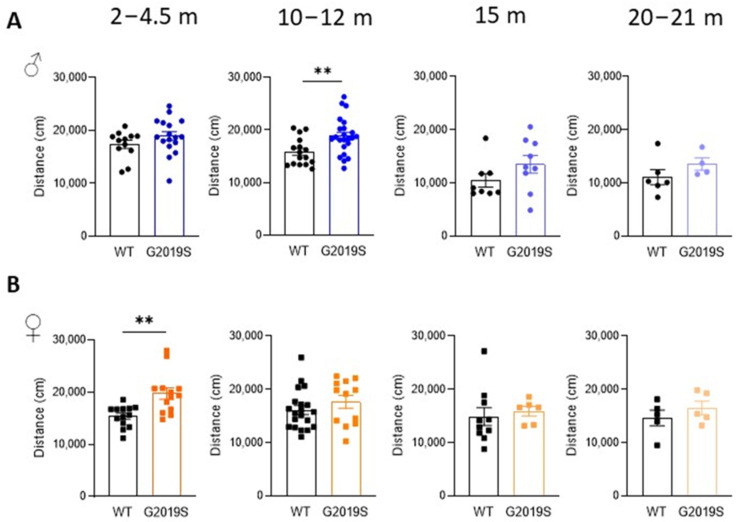
Male and female G2019S mice display hyperlocomotion at different ages. Total distance covered in the open field test during 60 min by male (**A**) and female (**B**) WT and G2019S mice at the ages indicated above the graphs (m: months). *N* = 4–23 mice in each group. ** *p* < 0.001 unpaired Student’s *t*-test. Circles represent data of male. Squares represent data of female. Colors indicate different ages.

**Figure 2 biomolecules-13-00051-f002:**
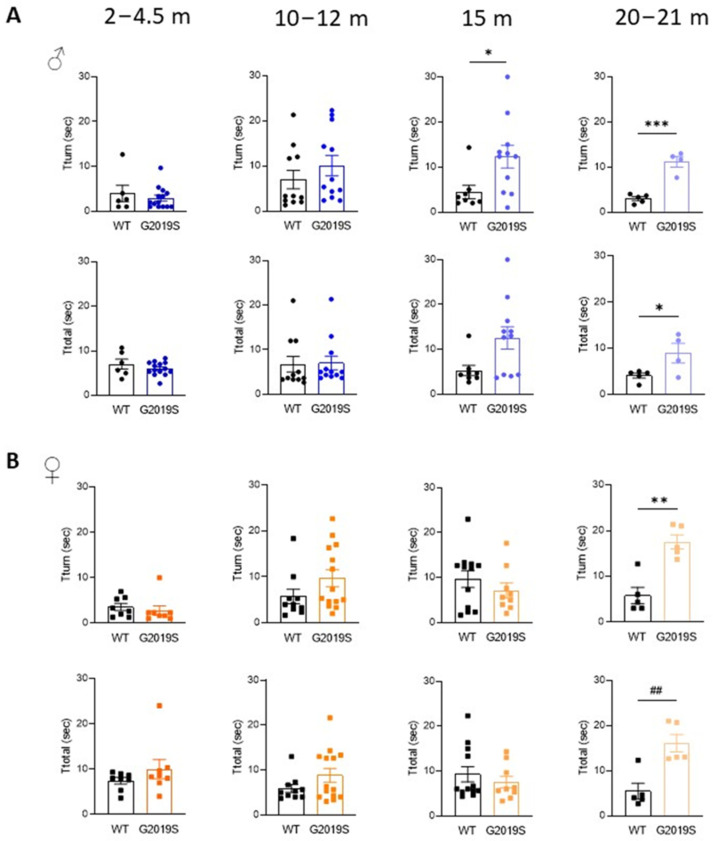
Male and female G2019S mice display fine motor impairment at different ages. Fine motor coordination was assessed with the pole test in male (**A**) and female (**B**) mice at four age ranges. Tturn: time taken by the mice to turn downward from the top of a vertical pole; Ttotal: total time to descend the pole. *N* = 4–14 mice in each group. * *p* < 0.05; ** *p* < 0.01; *** *p* < 0.001 unpaired Student’s *t*-test. ^##^
*p* < 0.01 Mann-Whitney U test.

**Figure 3 biomolecules-13-00051-f003:**
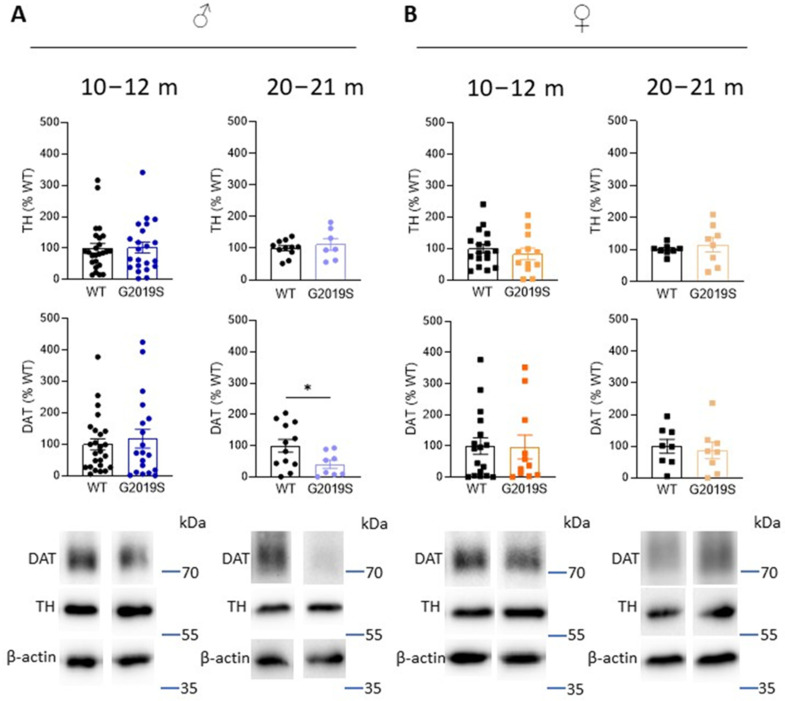
Old male G2019S mice display decreased DAT amount in the striatum. Western blotting of TH and DAT in the striatum of 10–12 months old and 20–21 months old male (**A**) and female (**B**) WT and G2019S mice. *N* = 4–25 mice in each group. * *p* < 0.05 unpaired Student’s *t*-test.

**Figure 4 biomolecules-13-00051-f004:**
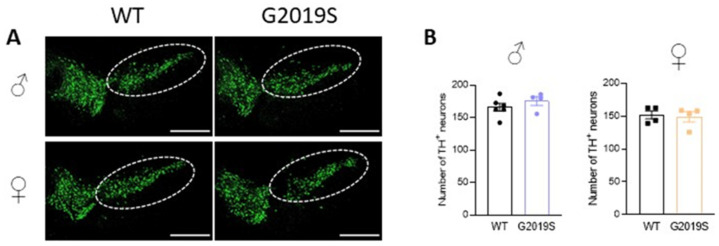
Intact cell counts in the SNc of old WT and G2019S male and female mice. (**A**) Confocal images showing immunofluorescence for TH in midbrain sections containing the SNc (dotted lines) of 20–21 months old WT and G2019S male and female mice; scale bars: 500 µm. (**B**) Number of TH-positive neurons in the SNc of 20–21 months old mice. *N* = 6 WT male mice, *N* = 4 G2019S male mice, *N* = 4 WT female mice, and *N* = 4 G2019S female mice.

## Data Availability

The data presented in this study are available on request from the corresponding author.

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
