# Peer review of "Age- and Sex-Dependent Behavioral and Neurochemical Alterations in hLRRK2-G2019S BAC Mice"

_biomolecules, 2022, doi:10.3390/biom13010051_

Round 1

Reviewer 1 Report

Authors assessed the behavioral dysfunctions and neurodegeneration that occur in male and female BAC LRRK2-hG2019S mice compared to their age-matched wild type littermates, at different age ranges, using behavioral tests, protein levels of TH and DAT, and immunohistochemistry of TH in the SNc neurons. The aim of study was to identify the effect of sex and age on the occurrence of motor and neurochemical deficits in BAC LRRK2-hG2019S mice. The authors need to explain some of the study results in the discussion section such as the mechanisms. To precisely answer this question in the BAC LRRK2-hG2019S model used in this study, it is suggested to perform transcriptomic and/or proteomic analyses of SNc neurons to reveal the mechanisms included in the PD pathogenesis in this model at different ages. This could be a new study to get enough samples for such analyses. At least the authors need to mention some mechanisms from literature which explain their results in the discussion section.

Materials and Methods

Line 109: Please detail the abbreviation of TH which is tyrosine hydroxylase then the abbreviation “TH” can be used in the following sections.

Line 144: Please correct “S.E.M” instead of s.e.m.

Results

The authors can share the videos of different behavioral tests via an online open access repository such as Figshare.

Discussion

Line 228: The authors need to discuss what the other possible mechanisms that might be involved in hyperlocomotion are. Similarly, the authors need to discuss what the mechanisms underlying motor deficits in old female G2019S mice and might differ from those in male mice are (Line 234).

Author Response

Materials and Methods

Line 109: Please detail the abbreviation of TH which is tyrosine hydroxylase then the abbreviation “TH” can be used in the following sections.

Line 144: Please correct “S.E.M” instead of s.e.m.

Reply to Reviewer’s comment:

We have now corrected these omissions/errors.

Results

The authors can share the videos of different behavioral tests via an online open access repository such as Figshare.

Reply to Reviewer’s comment:

We thank you Reviewer for this information and we will examine the possibility to share our videos in Figshare.

Discussion

Line 228: The authors need to discuss what the other possible mechanisms that might be involved in hyperlocomotion are. Similarly, the authors need to discuss what the mechanisms underlying motor deficits in old female G2019S mice and might differ from those in male mice are (Line 234).

Reply to Reviewer’s comment:

We have now included, in the Discussion section of our revised manuscript, possible mechanisms that could contribute to hyperlocomotion observed in G2019S mice as well as possible mechanisms that could underlie motor deficits in old female G2019S mice.

Reviewer 2 Report

This is an interesting study by Yao et al. describing age- and sex-dependent behavioral and neurochemical alterations in hLRRK2-G2019S BAC mice. Overall, this manuscript is well-written and meets the scope of biomolecules. But there are some concerns to be revised/improved in the manuscript.

1.       “m” should be defined as month.

2.       In figure 3, molecular weight from protein ladder should be indicated instead of the protein molecular weight. It would be better to view animal No. by plotting the graph with dots to represent each animal as other figures.

3.       In figure 4A, the images are too dim, and the contrast should be increased.

4.       in figure 4B, the TH+ neuron number is ~800 and unnormal. It’s better to normalize the No. to normal range.

5.       In addition, it would be great if authors could provide the survival curve. The authors mentioned that there is limited number of 20-21-month-old mice. I’m wondering if hG2019S affects mouse lifespan. Is it sex-dependent?

Author Response

  1. “m” should be defined as month.

Reply to Reviewer’s comment:

This is now defined in the legend of Figure 1.

  1. In figure 3, molecular weight from protein ladder should be indicated instead of the protein molecular weight. It would be better to view animal No. by plotting the graph with dots to represent each animal as other figures.

Reply to Reviewer’s comment:

We have modified Figure 3 according to our Reviewer’s suggestions.

  1. In figure 4A, the images are too dim, and the contrast should be increased.

Reply to Reviewer’s comment:

We have now increased the brightness of the images to show the labelling with higher visibility.

  1. in figure 4B, the TH+ neuron number is ~800 and unnormal. It’s better to normalize the No. to normal range.

Reply to Reviewer’s comment:

We thank our Reviewer for noticing this high number of TH+ neurons. We have corrected the values which are now expressed as an average of five sections in each animal.

  1. In addition, it would be great if authors could provide the survival curve. The authors mentioned that there is limited number of 20-21-month-old mice. I’m wondering if hG2019S affects mouse lifespan. Is it sex-dependent?

Reply to Reviewer’s comment:

The small number of 20-21 months old mice is not likely to be due to the G2019S mutation or the sex of the mice, but it is mostly due to the reduced availability of this age group. Indeed, we have sacrificed mice at two ages (10-12 and 20-21 months) to perform Western blotting experiments, and we have also sacrificed 10-12 months old mice for other projects. This explains why the 20-21 months old group has a lower number of mice. For this reason, we cannot establish survival curves from 2 to 21 months for individual mice.